# Regulation of mRNA Translation by Hormone Receptors in Breast and Prostate Cancer

**DOI:** 10.3390/cancers13133254

**Published:** 2021-06-29

**Authors:** Jianling Xie, Eric P. Kusnadi, Luc Furic, Luke A. Selth

**Affiliations:** 1Flinders Health and Medical Research Institute, Flinders University, Bedford Park, SA 5042, Australia; jianling.xie@flinders.edu.au; 2Translational Prostate Cancer Research, Peter MacCallum Cancer Centre, Melbourne, VIC 3000, Australia; Eric.Kusnadi@petermac.org; 3Cancer Program, Biomedicine Discovery Institute and Department of Anatomy and Developmental Biology, Monash University, Clayton, VIC 3800, Australia; 4Sir Peter MacCallum Department of Oncology, University of Melbourne, Parkville, VIC 3010, Australia; 5Freemasons Centre for Male Health and Wellbeing, Flinders University, Bedford Park, SA 5042, Australia; 6Adelaide Medical School, University of Adelaide, Adelaide, SA 5005, Australia

**Keywords:** androgen receptor, breast cancer, estrogen receptor, mRNA translation, mTOR, prostate cancer, protein synthesis

## Abstract

**Simple Summary:**

The estrogen and androgen receptors (ER, AR) are key oncogenic drivers and therapeutic targets in breast and prostate cancer, respectively. These receptors bind to DNA and regulate gene expression but emerging evidence indicates that they also play important roles in controlling the process of mRNA translation, which dictates cellular protein production. Here, we review the mechanisms by which abnormal activities of ER and AR can dysregulate mRNA translation in breast and prostate cancer cells. Specifically, we explore how the intricate cellular signalling pathways that keep mRNA translation in check are perturbed by aberrant ER and AR signalling, which can lead to enhanced cancer cell growth. We also discuss the potential of targeting mRNA translation as a strategy to treat patients with breast and prostate cancer.

**Abstract:**

Breast and prostate cancer are the second and third leading causes of death amongst all cancer types, respectively. Pathogenesis of these malignancies is characterised by dysregulation of sex hormone signalling pathways, mediated by the estrogen receptor-α (ER) in breast cancer and androgen receptor (AR) in prostate cancer. ER and AR are transcription factors whose aberrant function drives oncogenic transcriptional programs to promote cancer growth and progression. While ER/AR are known to stimulate cell growth and survival by modulating gene transcription, emerging findings indicate that their effects in neoplasia are also mediated by dysregulation of protein synthesis (i.e., mRNA translation). This suggests that ER/AR can coordinately perturb both transcriptional and translational programs, resulting in the establishment of proteomes that promote malignancy. In this review, we will discuss relatively understudied aspects of ER and AR activity in regulating protein synthesis as well as the potential of targeting mRNA translation in breast and prostate cancer.

## 1. Parallels between Aberrant Hormone Receptor Signalling in Breast and Prostate Cancer

Breasts and prostate are accessory sex organs that are exquisitely sensitive to sex hormones (i.e., estrogens and androgens). Malignancies arising from these organs are frequently driven by abnormal activity of the receptors of these sex hormones: indeed, >90% of prostate cancers (PC) are driven by the androgen receptor (AR) and ~70% of breast cancers (BC) are driven by the estrogen receptor-α (ER) [1]. As such, hormone deprivation therapies are the mainstay of treatment for locally progressive and advanced BC and PC. Although initially effective in most patients, these therapies are never curative and the resultant treatment-resistant tumours are aggressive and often fatal. Gaining a more complete understanding of the oncogenic activities of AR and ER is imperative for the development of more effective targeting strategies.

AR and ER are transcription factors that regulate the expression of genes involved in cancer development and progression, but also genes that are essential for the normal functioning of the prostate and breast glands. Both are members of the nuclear receptor superfamily and possess modular structures composed of N-terminal transactivation domains (NTD), DNA-binding domains (DBD), hinge regions and ligand binding domains (LBD) (Figure 1). Upon binding to their hormone ligands, AR/ER bind to chromatin at androgen/estrogen response elements and mediate coordinated recruitment of transcriptional co-regulators, resulting in activation or, more rarely, repression of gene expression [2,3]. In normal prostate and breast tissues, the transcriptional programs regulated by AR/ER in epithelial cells are responsible for promoting differentiation and regulating metabolism and the production of secreted proteins, whereas the AR/ER-regulated transcriptional programs in tumour cells promote cell growth and survival [4]. This switch in activities is mediated at least in part by altered expression of AR/ER co-regulator proteins and changes to the epigenome, resulting in altered AR/ER DNA binding profiles [5,6].

The similarities between AR/ER signalling in prostate and breast cancer extend to mechanisms of therapy resistance; indeed, most therapy-resistant cases of BC and PC exhibit alterations to the AR/ER signalling axes that enhance oncogenic signalling [1,7,8]. One key mechanism is mutation of the AR/ER LBD, which enhances the promiscuity of the receptor such that it can be activated by additional non-canonical ligands or even by antagonist drugs [7,8]. Other resistance mechanisms shared by AR/ER include the generation of constitutively active splice variants of the receptors, increased expression and/or activity of a plethora of transcriptional coactivators (e.g., steroid receptor coactivators, forkhead box protein A, GATA binding proteins, cAMP responsive element binding protein-binding protein/p300, switch/sucrose non-fermentable complex, E3 ubiquitin-protein ligases and steroid RNA activator, among others), and dampening of the activities of corepressors (e.g., nuclear receptor co-repressors, speckle type BTB/POZ protein and many others [9,10]. Collectively, these alterations to AR/ER signalling confer insensitivity to conventional hormone deprivation therapies (reviewed in [7,11]) and highlight the dependency of tumours on these pathways. 

## 2. Dysregulation of Translation Is a Common Feature of Cancer

### 2.1. Translation Initiation

Translation of mRNAs can be broadly divided into four steps: initiation, elongation, termination and ribosome recycling [12]. The initiation and elongation steps of translation are thought to be the most highly regulated and hence have been studied most intensively [13]. Dysregulation of translation in cancer, which occurs because of alterations in the levels or the activity of the components of translational machinery [14], can impact any of these steps. 

The major signalling pathways that confer translational control are summarised in Figure 2. A prominent feature of many cancers are mutations that lead to activation of PI3K (phosphatidylinositol-3-kinase)/PKB (protein kinase B) and the Ras (rat sarcoma)/MAPK (mitogen-activated protein kinase) pathways, including loss-of-function mutations of *PTEN* and gain-of-function mutations of PI3K and genes in the Ras/mitogen-activated protein kinase MAPK pathway [15,16,17,18,19,20]. PTEN (phosphatase and tensin homolog) is a phosphatase which dephosphorylates PtdIns(3,4,5)*P*3 (PIP3) and thus acts as a tumour suppressor to counteract PI3K activity. One major consequence of these mutations is increased activity of the mechanistic/mammalian target of rapamycin (mTOR) [21,22]. mTOR is a serine/threonine kinase that is part of two distinct complexes, mTOR complex 1 (mTORC1) and complex 2 (mTORC2) [23]. In response to a variety of stimuli including nutrients, oxygen availability, growth factors and hormones, mTORC1 stimulates protein synthesis by phosphorylating substrates such as the ribosomal protein (RP) S6 kinases (S6Ks; S6K 1 and 2 in humans) [24] and the eukaryotic translation initiation factor (eIF) 4E binding proteins (4EBPs; 4EBP1, 2 and 3 in humans). Upon mTORC1-mediated phosphorylation, 4EBPs are released from eIF4E, allowing it to form a heterotrimeric eIF4F complex—with the scaffold protein eIF4G and the RNA helicase eIF4A—that recruits mRNA to the ribosomes [25,26]. In addition, oncogenic activation of the RAS/MAPK pathway is thought to affect translation by stimulating phosphorylation of eIF4E via MAPK-interacting kinases (MNKs), as well as other translation initiation factors (e.g., eIF4B) and ribosomal proteins (e.g., rpS6) [27,28] (Figure 2). Targeting the subunits of the eIF4F complex has attracted substantial interest in BC/PC (e.g., see [29,30]) and will be further discussed below.

### 2.2. Translation Elongation

Notwithstanding that translation initiation is generally considered the rate limiting step of protein synthesis, more recent evidence shows that aberrant translation elongation may also play a prominent role in cancer [31]. In mammals, translation elongation rates are thought to be primarily regulated by an atypical calcium/calmodulin-dependent protein kinase called the eukaryotic translation elongation factor 2 (eEF2) kinase (eEF2K) [32]. eEF2K reduces elongation rates by phosphorylating and inhibiting eEF2 [33,34,35], a key translation factor that translocates nascent peptide chains from the A-site to the P-site of the ribosome in a GTP-dependent manner [36]. mTORC1 and MAPK suppress eEF2K, thereby increasing elongation rates [37,38] (Figure 2). In contrast, energy-sensing AMP-activating protein kinase (AMPK) phosphorylates and activates eEF2K via mTORC1-dependent and independent mechanisms to suppress translation and reduce energy consumption [39]. 

### 2.3. Integrated Stress Response (ISR)

In addition to the mTOR-dependent and -independent inhibitory effects of AMPK on protein synthesis, mTORC1 can be switched off in response to nutrient/energy deprivation or other stressors (e.g., hypoxia, acidosis) via the integrated stress response (ISR) pathway. The function of the ISR pathway is to restore cellular homeostasis or, in situations when its activation is prolonged, induce cell death. Intrinsic cellular stresses such as the unfolded protein response (UPR), which lead to endoplasmic reticulum (EnR) stress, can also trigger the ISR. Three arms of the UPR in mammals regulate distinct but highly orchestrated EnR-stress responses. The Protein Kinase RNA activated (PKR)-like EnR kinase (PERK)/eIF2α arm of the UPR reduces global protein synthesis to relieve EnR overload while inducing translation of a small subset of mRNAs, thus leading to transcriptional reprogramming that enhances EnR folding capacity. In parallel with this, the inositol requiring enzyme 1 (IRE1)/x-box binding protein 1 (XBP1) and the activating transcription factor 6 (ATF6) (and ATF6p50, a cleaved, active form of ATF6) arms of the UPR also contribute to the improvement of EnR folding capacity in 2 ways: (i) by altering transcription, and (ii) by triggering EnR-associated degradation (ERAD), which entails retro-translocation of misfolded proteins from the EnR to the cytosol and their subsequent degradation by the Ub/proteasome system (Figure 2). Additionally, EnR Ca^2+^ release through the inositol triphosphaste (IP_3_) receptor (IP_3_R) evokes cell fate decision events to either allow the cell to trigger or escape from apoptotic death (see [40,41] for reviews). 

The ISR arm of the UPR is centered on the phosphorylation of the α-subunit (eIF2α) of heterotrimer eIF2, which is catalysed by PERK [42] or three other kinases (general control nonderepressible 2 (GCN2), PKR and heme-regulated inhibitor (HRI)), that are activated by distinct stresses (amino acid deprivation, viral infection and heme deficiency, respectively) [43,44,45]. Phosphorylation of eIF2α inhibits eIF2B, which exchanges eIF2-GDP for eIF2-GTP and allows recycling of the ternary complex comprising eIF2, GTP and initiator methionyl-tRNA (tRNA_i_^Met^). The ensuing limitation in TC levels limits delivery of tRNA_i_^Met^ and halts global translation while allowing translation of a small subset of mRNAs characterized by inhibitory upstream open reading frames (uORFs) that encode transcription factors involved in stress response (e.g., ATF4, C/EBP homology protein (CHOP)), or are implicated in feedback loops that limit excessive ISR (e.g., GADD34 (growth arrest and DNA damage-inducible gene 34)) (reviewed in [46,47,48], also see Figure 2). Failure to do so ultimately leads to apoptotic cell death, which is exploitable as an anti-cancer strategy.

## 3. Genomic Alterations to Translation Factors in Breast and Prostate Cancer

Mutation of genes implicated in translation regulation and perturbation of translational programs appears to be a common feature of BC/PC, which provides a strong rationale for targeting translational machinery in these diseases. In particular, mutations in genes encoding proteins upstream of mTORC1, including core components of the PI3K/AKT pathway (i.e., *PIK3CA*, which encodes the p110α subunit of PI3K, *PTEN* and *AKT1*) and the MAPK pathway (*MAP3K1*, *MAP2K4*, *BRAF* and *HRAS*) are frequent in BC and PC and considered to be drivers of tumourigenesis [49,50,51]. Mutations in the *MTOR* gene itself are also present in BC (1.85%) and PC (0.4%) patients [52,53,54] (Table 1). 

Notably, cancer-related alterations in translational machinery and associated factors do not uniformly affect translation of all cellular mRNAs but rather cause selective reprograming of the translatome to favour synthesis of factors that promote tumorigenesis, tumor progression and drug resistance [23,55]. These differences in translational programs of normal and neoplastic cells could yield a therapeutic window to selectively eradicate cancer cells by targeting translational machinery, while causing minimal toxicity [23]. This concept has spurred recent efforts to design and repurpose translational inhibitors and apply them in oncological indications [14]. Herein, we review recent findings suggesting that dysregulation of the abovementioned mechanisms of translational regulation may play a pivotal role in BC and PC, with a particular focus on the connections between perturbations in mRNA translation, aberrant ER and AR signalling and potential therapeutic applications. 

## 4. Interplay between mRNA Translation and ER Signalling 

### 4.1. ER Selectively Controls Translation Initiation 

Early evidence of mRNA translation being regulated by estrogen was provided in the late 1950s and the early 1960s [56,57,58,59,60]. More specifically, estrogen stimulated the incorporation of glycine-2-C^14^, used as a gross measure of protein synthesis, into rat and chicken cells. However, the effect of estrogen on protein synthesis in BC cells was not rigorously explored until the beginning of the 21st century. For example, an early study found that estrogen facilitated the association of mRNAs with translationally active polyribosomes [61], implying that not all effects of estrogen on BC cell functions were through de novo mRNA synthesis. More recent work is beginning to shed mechanistic insight into how ER regulates mRNA translation. A role for ER in translation appears to relate to its ability to localize to the plasma membrane within lipid rafts (i.e., caveolae)-enriched regions [62], which can be regulated by direct palmitoylation (at C447/C451 in human/mouse respectively) of the ER [63]. There, ER is thought to transactivate tyrosine kinase receptors including epidermal growth factor receptor (EGFR) and insulin-like growth factor 1 (IGF1) receptor (IGF1R) via Src [64,65,66], or to associate with hematopoietic PBX-interacting protein thus facilitating recruitment of Src to the p85 regulatory subunit of PI3K [67]. In this manner, ER is thought to activate PI3K/PKB and MAPK pathways and thus impact translation. Such extra-nuclear functions of nuclear receptors have been proposed to allow more rapid responses than their transcriptional effects [68]. In the context of cap-independent translation, a shorter ER isoform termed ERα46 is translated via an IRES (internal ribosomal entry site)-dependent mechanism in response to EnR stress [69]. However, it is still unclear whether ER has a role in regulating the translation of cancer-associated IRES-dependent mRNAs (reviewed in REF e.g., [70]) such as hypoxia-inducible factor 1α (HIF1α) or the anti-apoptotic protein B-cell lymphoma 2 (BCL2).

Whole-genome/exome sequencing studies revealed that mTORC1 signalling is one of the most upregulated pathways in metastatic BC cells compared to paired primary tumours [71,72,73,74]. The mTOR complexes play a central role in converging upstream signals evoked by ER to control translation initiation. 17β-estradiol (E2, a classic ER agonist) treatment could robustly evoke the phosphorylation of mTORC1 (S6K1 and 4EBP1) and mTORC2 (PKB) downstream targets within 5 min in MCF-7 cells [75]. Analysis of transcriptomic data revealed a strong correlation between ER and mTORC1/2 signalling-regulated genes at the level of gene expression [76]. Indeed, BHPI (3,3-bis(4-hydroxyphenyl)-7-methyl-1,3-dihydro-2H-indol-2-one), a non-competitive ER biomodulator, was able to induce S6K phosphorylation (a readout of mTORC1 activity) and an inhibitor of mTORC1 (rapamycin) reversed the stimulatory effect of E2 on protein synthesis [77]. A recent phosphoproteomics study identified a range of mTORC1 phosphorylation sites from its classic substrates (e.g., S6K, eIF4B, proline-rich Akt substrate of 40 kDa (PRAS40) and La-related protein 1 (LARP1), among others) that could be induced by E2 and are sensitive to rapamycin treatment in MCF-7 cells [78]. Importantly, the authors identified *DEPDC6*, which encodes for DEPTOR (DEP-domain containing mTOR-interacting protein), as a novel ER-target gene. DEPTOR is a well-established binding partner of mTORC1/2 which negatively regulates the activity of both complexes [79], and thus serves as a tumour suppressor against PC progression [80]. Therefore, the induction of *DEPDC6* expression is more likely to act as a negative feedback regulatory mechanism in response to E2-induced mTORC1 activation (Figure 3), but how E2/ER stimulates mTORC1 remains to be elucidated. One obvious possibility is that E2/ER achieves this through the activation of signalling pathways (e.g., PI3K/MAPK) upstream of mTORC1 [81]; alternatively and/or additionally, ER could activate mTORC1 via a non-genomic mechanism as mentioned above. Moreover, ER is a known mTORC1 substrate (see Section 4.2), and therefore phosphorylated/active ER could potentially trigger positive feedback loops to further stimulate mTORC1; such feedback loops are known to be commonly driven by mTORC1 substrates [82,83]. 

In addition, mTORC1 activation by E2/ER also facilitated the association between the octameric eIF3 and eIF4F, which is required for the formation of 48S preinitiation complex during mammalian translation initiation, as this effect can be reversed by rapamycin [84]. Notably, expression levels of most eIF3 subunits are upregulated in cancer, except for eIF3e and f, which are downregulated in tumour cells and may act as tumour suppressors [85]. *eIF3f* transcription was greatly reduced upon E2 treatment in MCF-7 cells. However, E2 has minimal effect on eIF3f protein expression, and quite unexpectedly, siRNA-mediated knockdown of ER reduced eIF3f protein levels, implying that the translation of *eIF3f* was actually enhanced by the presence of ER [85]. How ER can contribute to the synthesis of eIF3f is still unclear; it may involve the activation of mTORC1, since its substrate S6K1 has been shown to interact with and phosphorylate eIF3 [86,87].

Perturbations in the transcription profile may trigger regulatory mechanisms which attempt to compensate for or buffer those changes at the level of mRNA translation, a phenomenon known as translational “offsetting” [88]. Analogous to this, one would expect that altering ER’s transcriptional activity with ER modulators or EDT will induce this adaptive response to maintain translational homeostasis in tumour cells. Analysis of polyribosome-associated mRNAs in BM67 cells (a cell line derived from cPTEN^fl/fl^ mice, which was obtained by crossing PB-Cre4 with PTEN^flox/flox^ mice) stably expressing shRNAs against ER (shER) revealed that ER depletion triggered translational offsetting [89]. Of importance, mRNAs that were downregulated but translationally offset in shER BM67 frequently contained short 5′-untranslated regions (UTRs) and lacked miRNA target sites at their 3′-UTRs [89]. This is of particular interest since ER is known to either up- or downregulate the expression of several miRNAs in order to fine-tune the translatome as another mechanism to promote BC tumour growth [90,91,92]. In contrast, mRNAs that were induced by ER depletion while being translationally offset are enriched in a specific set of codons decoded by tRNAs bearing a modified uridine (5-methoxycarbonyl-methyl-2-thiouridine) at position 34 (mcm^5^s^2^U_34_) [89], which is formed upon a sequential catalysis by ELP3 (elongator complex protein 3, to generate cm^5^U), ALKBH8 (AlkB homolog 8, to generate mcm^5^U) and CTU1/2 (cytoplasmic tRNA 2-thiolation protein 1/2, to generate mcm^5^s^2^U). These mcm^5^s^2^U_34_-modifying enzymes, particularly ELP3, have recently been linked to therapy resistance in BRAF^V600E^ driven melanoma [93], BC [94] and colorectal cancer [95]. Importantly, *Elp3* mRNA association with polysomes and ELP3 protein expression level was greatly reduced in shER BM67 cells as well as other tested ER-negative cell lines [89], implying that ER may also indirectly impact translation elongation of mRNAs enriched in codons containing mcm^5^s^2^U_34_ tRNAs (Figure 3).

### 4.2. mTORC1 Directly Controls ER-Mediated Transcription

Direct modification of ER by mTORC1 represents another important link between these factors. ER possesses a highly conserved TOR signalling motif (FPATV, Figure 1), which typically starts with a phenylalanine and is present in most of the mTORC1 substrates [96]. Indeed, mTORC1 can directly phosphorylate ER on Ser104/Ser106, resulting in the enhancement of ER’s transcriptional activity amongst its target genes [97]. This potentially provides a positive feedback link by which ER stimulates a central modulator of protein synthesis to further strengthen its function as a transcription factor (Figure 3). This finding can be substantiated by the fact that, as evidenced by chromatin immunoprecipitation analysis of ER’s binding to the estrogen response element promoter on the pS2 gene, rapamycin was inhibitory to ER’s transcriptional activity in MCF-7 cells [98].

A high level of mTORC1 activity is a predictor of poor progression-free survival (PFS) outcomes in ER^+^/PR^+^ BC patients [99]. However, a recent study from Rutkovsky et al. found that 4EBP1, a well-established negative regulator of mTORC1 activity [100], is highly phosphorylated and overexpressed in a range of metastatic ER^+^ BC cell lines harbouring the 8p11-p12 amplicon, and contributes to BC cell proliferation [101]. DNA amplification of the 8p11-p12 genomic region, which contains the gene encoding 4EBP1 (*EIF4EBP1*), is frequent in endocrine resistant BC. Knocking down 4EBP1 with lentiviral shRNA reduced the replication of BC cells harbouring the 8p11-p12 amplicon and, unexpectedly, repressed the level of ER protein expression [101]. Hyperphosphorylated 4EBP1 is resistant to ubiquitin-mediated degradation and thus highly stable [102]. Interestingly, knocking down eIF4E can deplete 4E-BP4EBP1 from BC cells [102]; therefore, in addition to amplification of the *EIF4EBP1* gene, the observed increase in it therefore suggests it is plausible that increased levels of 4EBP1 in ER^+^/8p11-p12^+^ BC cells may also be a result of elevated eIF4E [102]. As further evidence of the relevance of 4EBP1 in breast cancer, genomic profiling studies have identified gene fusions such as *TACC1-EIF4EBP1* [103] which are thought to enhance its activity by promoting 4EBP1 autophosphorylation [104,105].

### 4.3. ER Is an Important Player in the UPR

Translation initiation can be activated by anabolic signals from environmental cues favouring cell growth, whereas it is essential for the process to be reversed when cells are facing stressors. In this respect, UPR plays a crucial role in ensuring cell survival by switching off translation initiation in response to stressors. It has been shown that ER can also regulate UPR under catabolic conditions. One of the earliest examples of how ER ligands induce UPR was from Shapiro’s group, who discovered that that the ER inhibitor BHPI rapidly reduced protein synthesis rates in three types of ER^+^ BC cells (MCF-7, T47D and BG-1/MCF-7), concomitant with an induction of PERK/eIF2α and AMPK/eEF2 phosphorylation, indicative of EnR stress/UPR and the attenuation of translation elongation respectively (Figure 3). Conversely, treating the MCF-7 cells with E2 increased global protein synthesis by approximately 3-fold [77]. The authors further showed that the PERK inhibitor GSK2606414 was able to enhance global protein synthesis to up to 6-fold in BHPI-co-treated MCF-7/T47D cells, whereas knocking down PERK with siRNA strikingly decreased protein synthesis in untreated cells to less than 10% within 24 h, implying that EnR activation was responsible for the reduced protein synthesis in response to BHPI treatment [77]. However, it is still unknown whether the stimulation of the AMPK/eEF2K pathway also contributes to BHPI-mediated protein synthesis inhibition. A more recent study also found that E2 treatment could enhance protein synthesis in MCF-7:5C cells, which are aromatase inhibitor (AI)-resistant MCF-7 cells generated by long-term estrogen deprivation. However, in this model, the effect of E2 was much weaker than was reported in AI-sensitive cell line models [77], implying that estrogen depletion desensitises the cells from ER agonist-induced protein synthesis. 

In spite of increasing evidence that ER can modulate the UPR, the mechanism by which it achieves this remains unclear. One possibility is through the regulation of intracellular Ca^2+^ channels such as IP_3_R and SERCA, which are Ca^2+^-ATPases that transport cytoplasmic Ca^2+^ into the EnR and thus act as secondary messengers for EnR stress/UPR induction (reviewed in [47]). For instance, BHPI can evoke the activation of phospholipase Cγ (PLCγ), which results in elevated IP_3_ production and subsequently increases intracellular Ca^2+^ levels by depleting Ca^2+^ from the EnR reservoir [77] (Figure 3). Whether ER’s ability to activate PLCγ is due to direct regulation of its gene, similar to other UPR-related genes [92], is unknown.

## 5. New Roles for AR Signalling in mRNA Translation

### 5.1. AR Indirectly Regulates Translation Initiation

Similar to estrogenic control of protein synthesis [56,57,58,59,60], a central role for androgens in stimulation of protein synthesis in the prostate was also proposed over half a century ago [106]. Initial observations suggested that androgen-dependent induction of protein synthesis in prostate is rapid [107,108,109,110] and thus unlikely to stem only from the effects of AR on transcription. However, the mechanisms underlying AR translational reprogramming in PC are incompletely understood. Highlighting this issue, a recent study found that de novo protein synthesis was strikingly enhanced in castrated cPTEN^fl/fl^ mice, which apparently contradicts early historical findings [30]. Enhanced protein synthesis in this model was correlated with a decrease in 4EBP1 levels and concomitant increase in eIF4F complex assembly [30]. Low levels or complete loss of AR also coincided with reduced 4EBP1 expression in AR-low/null human AR program-independent prostate cancer cell line and metastatic castration-resistant PC (CRPC) LuCaP patient-derived xenografts (PDXs). Consistently, AR-low PC cells exhibited higher sensitivity to 4EBP1 over-expression as compared to corresponding models with high levels of AR [30]. As such, these findings suggest that the absence of AR can also promote protein synthesis, and that the effects of AR on translation may be mediated via regulation of 4EBP1/eIF4F. It has also been shown that AR inhibition by bicalutamide in metastatic CRPC PDXs promoted eIF4E phosphorylation, and sensitized the PC cells to MNK inhibition [111]. Taken together, these reports support the notion that the components of eIF4F may represent a druggable target for AR-low PC.

In addition to affecting 4EBP1 levels, AR may also regulate 4EBP1 activity via modulating mTORC1 signaling. For instance, androgen treatment induced expression of L-type amino acid transporters 3 (LAT3) in LNCaP (PC cell line carrying a AR-FL^T877A^ mutation) cells, whereas bicalutamide reversed these effects [112]. *LAT3* is highly expressed in primary prostate tumours [113,114,115,116] and depleting LAT3 effectively diminished prostate cancer cell proliferation in vitro [112]. AR can also regulate mTORC1 indirectly via a signalling cascade involving kallikrein related peptidase 4 (KLK4), promyelocytic leukemia zinc finger (PLZF) and regulated in development and DNA damage responses 1 (REDD1) [117] (Figure 4). Association of the AR-regulated factor KLK4 [118] leads to destabilization of PLZF and REDD1, relieving mTORC1 from REDD1-mediated inhibition. In other words, one would expect that the ablation of AR inhibits mTORC1, but this is seemingly contradictory to the fact that AR-low PC cells have a high protein synthesis rate [30]. One explanation can be that low levels of 4EBP1 expression in AR-low/null cells render eIF4E insensitive to mTORC1 inhibition [30]; hence the eIF4F complex becomes “constitutively active”.

The effect of AR signalling on mTORC1 activity appears to be context-dependent. In contrast to the findings described above, Zhang et al. recently demonstrated that knockdown of AR can trigger mTORC1 stimulation in hepatocellular carcinoma (HCC) cells MHCC-97L, as a result of downregulation of the classic AR target, FK506 binding protein 5 (FKBP5), which acts as an AKT/mTORC1 upstream inhibitor [119]. In addition, although rapamycin treatment is known to enhance AR’s transcriptional activity in PC cells [120], rapamycin blunted the transcriptional activity of AR in HCC (SNU423 and MHCC-97L) cells. Therefore, it appears that either the stimulation (PC) or the depletion (HCC) of AR can activate mTORC1; similarly, mTOR inhibition can either stimulate (PC) or repress (HCC) transcriptional activity of the AR. This apparent paradox may be reconciled as a cell type-specific phenomenon but warrants further investigation. One possibility is that long-term rapamycin treatment can suppress mTORC1 and mTORC2 in some cell lines (e.g., LNCaP and PC3 [120,121]), but not others [121], and therefore the effect of mTOR inhibition may depend on its effect on mTORC2. The use of an ATP-competitive mTOR inhibitor (mTORi) that strongly suppresses the activation of both mTORC1 and 2 ([122], also see Section 6 below), or mTOR complex-specific genetic ablation, may help to dissect the precise role of the regulation of AR by mTOR. 

Beyond control of translation factors, other mechanisms of interplay between AR and mTOR have been documented. For example, it was found that AR and mTOR can interact on chromatin, and co-regulate the transcription of genes implicated in metabolic rewiring in PC cells [123,124,125]. Additionally, AR/mTOR can also promote the cleavage and hence the nuclear translocation and activation of sterol regulatory element-binding transcription factor 1 to induce the expression of crucial lipogenic genes such as *FASN* and *SCD1* [125]. As such, it is tempting to speculate that by targeting both AR and mTOR one may even “kill two birds with one stone”—simultaneously modulating oncogenic translational and transcriptional activities within a PC cell.

In comparison to cap-dependent translation, how AR regulates IRES-dependent/cap-independent translation is less well understood. A dramatic increase in IRES signals was observed in the testis of mice treated with testosterone and siRNA-based silencing of AR reduced IRES signals in vivo [126], suggesting that AR activation promotes IRES-dependent translation. However, in another study, AR inhibition with bicalutamide was also shown to promote IRES-dependent translation in CRPC cells [111]. In short, more work is required to precisely elucidate the role of AR in IRES-dependent/cap-independent translation.

### 5.2. AR and UPR

Multiple lines of evidence link the AR and UPR pathways in PC. Early work found that AR expression is positively correlated with several UPR-related genes in prostate tumours [127], which may relate to the fact that transcription of some UPR-related genes (e.g., *NDRG1* and *HERPUD1*) is responsive to androgen treatment in PC cells [128]. This work suggested that AR stimulates UPR, which was confirmed by the finding that the synthetic androgen R1881 induces the IRE1/XBP1 arm of UPR in LNCaP cells (Figure 4) [129]. Conversely, silencing AR in LNCaP cells reversed the induction of *ERN1* transcription and reduced the levels of spliced XBP1 [127,130]. Importantly, XBP1 was found to be highly expressed in PC patient tumours compared to benign tissues [127]. The synthetic androgen also induces ERAD in several PC cell lines, and high expression of ERAD-related genes (*gp78*, *Hrd1* and *SVIP*) has been observed in PC patient tissues [131]. Additionally, Yang et al. found that DHT-induced AR protein expression in murine embryonic stem cells (mESCs) correlated with induction of all three arms of UPR, evidenced by increased *XBP1* mRNA splicing, elevated phosphorylation of PERK, upregulated levels of mRNA encoding CHOP mRNA and ATF6p50 proteins [132]. Sheng et al., however, reported only modest increases in mRNA levels of ATF6/ATF6p50 in response to R1881, implying a weaker activation of the ATF6 arm of UPR in comparison to the IRE1α/XBP1 arm. ATF4 and CHOP levels were also shown to be elevated in this study, although surprisingly the levels of phosphorylated PERK/eIF2α and total PERK were markedly decreased [127]. These apparent discrepancies between the studies indicate that the extent to which AR-mediated regulation of eIF4F complex assembly [30] and PERK/eIF2α phosphorylation/ISR [127] contributes to translational perturbations in prostate cancer remains to be established. Nevertheless, the overall body of evidence demonstrates that AR (activation) induces UPR, which can lead to apoptotic death in some non-PC cells (e.g., mESCs and ovarian cells) [132,133]. The reader is also directed to an excellent recent review [134] which further covers this topic.

## 6. Targeting Pathways That Regulate mRNA Translation in BC and PC

### 6.1. Targeting Pathways That Regulate mRNA Translation

#### 6.1.1. PI3K

Most of the *PI3KCA* mutants in BC are oncogenic (promote growth, proliferation and survival of BC cells) and enhance the activity of the PI3K pathway, particularly in response to ligands such as insulin and IGFs [135,136]. Accordingly, depletion of PI3KCA and PI3KCB is synergistic with estrogen deprivation in ER^+^ BC cells with *PI3KCA* mutations and/or *PI3KCB* amplification [137]. Dual PI3K/mTORis (BKM120, BYL719, RAD001 and BEZ235) also exert strong anti-proliferative effects on a range of ER^+^/human epidermal growth factor receptor 2 (HER2)^+^ BC cell lines carrying *PI3KCA* mutations both in vitro and in vivo [138]. Paradoxically, *PI3KCA* mutations are favourable prognostic markers of PFS [139,140] for ER^+^/HER2^+^ BC patients, and despite high levels of PI3K/AKT signalling, they appear to be associated with reduced mTORC1 activity [140]. ER^+^/HER2^+^ BC patients harbouring *PI3KCA* mutations are also more responsive to tamoxifen (a selective ER modulator) monotherapy [140]. This can be explained in part by elevated levels of genes that negatively regulate mTORC1 such as *PP2A* and *PML* [140], feedback mechanisms in response to PI3K activation (see [82,83]) and/or mutant-specific signalling patterns. Moreover, a recent study found that PDXs from the luminal AR^+^ subtype (LAR^+^) of ER^−^/PR^−^/HER2^−^ BC (TNBC), which accounts for up to 9% of all TNBCs [141] and is characterized by AR activation, are highly enriched in *PI3KCA* and *AKT1* mutations (100% in LAR^+^ TNBC vs. 7.5% in other subtypes of TNBCs) [142]. Enzalutamide-resistant LAR^+^ TNBC PDXs are remarkedly sensitive to PI3K, mTOR or PI3K-mTOR dual inhibitors [142]. PI3K and MAPK pathway inhibitors have been extensively studied as anti-BC/PC agents and interested readers are referred to several excellent reviews on this topic [143,144,145,146]. 

#### 6.1.2. MNK

MNKs are promising therapeutic targets within the translational control signalling network. Targeting MNKs may be a safer option since they are unessential to normal cell physiology, as evidenced by the fact that MNK1/2 double knockout mice are viable and fertile [50]. Interestingly, Njar’s group discovered that a group of compounds, the C-4 heteroaryl retinamides, could simultaneously provoke the degradation of AR-FL, AR-V7 (AR splice variant 7, a variant that lacks the ligand-binding domain and hence can signal in the absence of androgen) and MNK1/2 [147,148]. Retinamides exhibited substantial anti-proliferative, anti-migratory/invasive (in vitro) and anti-growth (xenograft) properties in a range of AR-positve and -negative PC cell lines [147,148], suggesting that simultaneously inhibiting the AR and the MNKs can potentially be a new strategy against AR^+^ PC; this also implies that the MNK inhibitors (MNKis) may be useful in an AR-negative context. In support of this notion, it has been demonstrated that AR serves as a suppressor of eIF4E phosphorylation [111], which is catalysed only by MNKs [50], and thus loss of AR would evoke MNK activation. On the other hand, mTORis may also reactivate MNK2 by alleviating it from inhibitory post-translational modifications (phosphorylation at Ser74 and Ser437) catalysed by mTORC1 [149,150]. Indeed, in a PC patient tissue microarray analysis, MNK2^Ser74^ was less phosphorylated in tumour specimens with a high Gleason score [149]. In support of these observations, the MNKi CGP57380 can overcome the resistance to mTORis in PC cells [111]. A monotherapy regimen with a more specific MNKi, eFT508 [151], completed a CRPC-related phase II clinical trial (NCT03690141) in April 2021.

Similarly, tamoxifen-resistant ER^+^ BC cells also exhibited elevated mTORC1 activity, concomitant with high levels of total and phosphorylated eIF4E; suppression of mTORC1/MNK1 activity using everolimus/CGP57380, or reducing eIF4E levels via si/shRNA, resensitized cells to tamoxifen [152]. CGP57380 exerts some off-target effects on other kinases (CK1, BRSK2 and MKK1) [153], a cautionary note when interpreting results with this agent. 

#### 6.1.3. eIF4E

Because inhibition of global translation will unavoidably disrupt physiological functions of the cell, one can expect that future therapeutic strategies aiming at mRNA translation will tend to be more selective towards specific sets of targets. Most of the factors implicated in translation regulation are essential for normal cell function, which is a major hurdle for any treatment strategy targeting the translation machinery. However, eIF4E may serve as a key translation target for anti-cancer treatment. A whole body haplo-insufficient eIF4E^+/−^ genetically engineered mouse model (GEMM) revealed that a “full dosage” of eIF4E is not required for normal development, yet it aids the translation of specific subsets of mRNAs, particularly those that regulate the production of reactive oxygen species, to promote tumour progression [154]. In addition, genetic ablation of the MNKs, the only in vivo eIF4E kinases, did not affect murine viability or fertility [27]. Importantly, eIF4E is highly phosphorylated in advanced PC patient biopsies [149,155]. Knocking-in mutation of eIF4E^Ser209^ (MNKs’ phosphorylation site) to alanine (eIF4E^S209A/S209A^) in mouse embryonic fibroblasts conferred them resistance to oncogenic transformation [155]. Using the cPTEN^fl/fl^ GEMM, it has also been shown that cPTEN^fl/fl^ × eIF4E^S209A/S209A^ knock-in mice are resistant to the loss of PTEN-induced PC development [155]. Moreover, Hsieh et al. [156] reported that INK128, an ATP-competitive mTORi (see Section 6 below) that blocks the activities of both mTORC1 and 2, exerted strong cytotoxicity in and effectively prevented the metastasis of PC cells in cPTEN^fl/fl^ mice [156]. Collectively, these studies serve as a basis for future research on the potential of targeting eIF4E in PC.

#### 6.1.4. eIF4A

An alternative therapeutic strategy to targeting upstream signalling pathways is direct interference with the components of the eIF4F complex. A study from Modelska et al. [38] has revealed that high levels of eIF4A1, eIF4B and eIF4E are all linked to unfavourable clinical outcomes for ER^−^ BC patients, whereas the tumour suppressor programmed cell death 4 (PDCD4), which binds to and inhibits eIF4A [157], is positively correlated with favourable outcomes in ER^+^ BC patients. Knock-down of eIF4A or eIF4B using siRNA or over-expressing PDCD4 effectively diminished MCF-7 cell (an ER^+^/PR^+^ BC cell line) proliferation in vitro [38]. Investigation into drugs that specifically target the eIFs is still at an early stage, although some have showed promising results in preclinical studies in BC models [158,159]. For instance, eIF4A inhibitors (eIF4Ais), which are derivatives of a class of natural products called the flavaglines, were shown to have strong anti-tumour actions against BC/PC cells both in vitro and in vivo [158,159,160,161,162,163,164,165]. Most of the studies related to eIF4Ais are still at a preclinical stage; however, a newly synthesized flavagline derivative (eFT226 [166]) has just entered phase I/II clinical trials (NCT04092673) against selected advanced solid tumours. Interestingly, targeting eIF4A influenced the BC transcriptome and translatome in a manner distinct from mTORi-mediated eIF4E inhibition [167], indicating that eIF4A and eIF4E may affect different subsets of messages in cancer cells and suggesting that concomitant targeting of eIF4A and eIF4E is a plausible combinatorial therapeutic strategy. 

### 6.2. Targeting Ribosome Biogenesis

Eukaryotic ribosomes, comprised of ~80 RPs and 4 ribosomal RNAs, are macromolecular complexes that function to synthesize proteins from mRNAs. Using a genome-wide CRISPR screen (over 70,000 single guide RNAs) coupled with patient-derived metastatic BC cells, Ebright et al. [168] recently discovered that ribosomal genes, especially *eL15*, *uL29* and *eL13* (encoding RPL15, RPL35 and RPL13, respectively) were highly enriched in circulating tumour cells (CTCs). In particular, over-expression of RPL15 in CTCs promoted global protein synthesis and metastasis in the lung in mice. In addition, patient CTCs with the highest expression levels of RPs were associated with the worst clinical outcomes. Other factors that were highly correlated with poor patient survival rates included genes that encode for the eIFs and mTOR. A combinational therapy using omacetaxine, an FDA-approved translational elongation inhibitor, and palbociclib, a CDK4/6 inhibitor approved for BC treatment, effectively prevented the metastasis of RPL15-enriched CTCs [168]. Similar to these findings in BC, Rebello et al. [169] found that the combination of two drugs (CX-5461 and CX-6258), two orally available agents which inhibit RNA polymerase I transcription and the PIM kinase respectively, exerted strong anti-proliferative and pro-apoptotic effects on both high c-Myc and PTEN-null murine prostate tumour models. A later study demonstrated that the CX-5461/CX-6258 combination therapy effectively inhibited growth, increased DNA damage responses and suppressed mTOR signalling in four distinct PC PDX models obtained from CRPC patients resistant to second-generation ADT agents, including abiraterone and/or enzalutamide [170]. Taken together, these studies highlighted a promising future of targeting the translation machinery, especially the ribosomes, for BC/PC treatments.

### 6.3. Targeting the UPR 

To cope with the abnormally high protein synthesis demand under a nutrient deprived, hypoxic and acidotic microenvironment, it is critical for cancer cells to maintain energetic balance and proteostasis, for example, by triggering ISR or activating AMPK. Nguyen et al. recently discovered that, unexpectedly, the co-existence of PTEN deletion and high c-Myc expression (cPTEN^fl/fl^ × Myc^Tg^) in murine prostates resulted in dampening of the increase in global protein synthesis (compared to PTEN-null or high c-Myc alone), which occurred concomitantly with an induction of phosphorylated PERK, one of the four eIF2α kinases [42]. In this study, the ablation of PERK rescued the reduction in protein synthesis in cPTEN^fl/fl^ × Myc^Tg^ mice [171]. As such, one can expect that ISR inhibition may lead to devastating consequences for PC cells but have a limited toxic effect on normal cells. Indeed, the PERK inhibitor ISRIB [172] induced apoptosis in PC cells, reduced tumour volume, and prolonged the survival rates of cPTEN^fl/fl^ × Myc^Tg^ mice, as well as of mice bearing metastatic CRPC PDXs [171]. However, although alleviation of ISR exerted strong anti-tumour effects in the cPTEN^fl/fl^ × Myc^Tg^ GEMM [171], it is not clear whether the dampening of protein synthesis rates in those mice reflects the induction of ISR.

An alternative therapeutic strategy for PC would be to target the IRE1/XBP1-axis of UPR. Toyocamycin, an IRE1α inhibitor, effectively diminished LNCaP or VCaP xenograft growth in vivo [127]. Similarly, MKC8866, another specific inhibitor to IRE1α RNase activities, potentiated the activity of conventional AR-targeted therapies (enzalutamide, abiraterone and cabazitaxel) in mice bearing VCaP xenografts [129]. Taken together, these studies [127,129] thus provide a molecular basis for targeting the IRE1/XBP1 arm of UPR for PC treatment. 

## 7. Drawbacks and New Hopes in Treatments Targeting mTOR in BC and PC

### 7.1. Rationale for Application of mTORis as an Anti-BC/PC Treatment

As described above, there are various potentially druggable targets within the translation machinery that can be exploited for anti-BC/PC therapies. Among them, mTORis are one of the most promising strategies. This is mainly due to a suite of unique features associated with mTOR. Firstly, mTOR is a protein kinase initially discovered to be the sole target of a highly specific natural compound, rapamycin, whereas most of the early identified kinase inhibitors possess severe off-target effects [153,173,174]. Secondly, targeting upstream signalling components such as PI3K, AKT, Raf, MEK, etc., will unavoidably trigger stronger side effects since they also regulate other essential cell processes, whereas mTOR acts downstream of these signalling pathways and its best established role so far is the homeostatic control of protein synthesis. It has been shown that the ratio of eIF4E to 4EBPs correlates with response to mTORC active site kinase inhibitors [175]. Lastly, mTORis, especially rapamycin analogues (rapalogs), have been in development for >20 years; indeed, some have been approved by the FDA for clinical use against certain types of cancer (e.g., kidney, renal and pancreatic) since the early 2000s.

### 7.2. Early Trials Using Rapalogs in BC/PC

Rapalogs were initially successful in early clinical trials against BC [176,177,178], as evidenced by reasonable anti-tumour activities, well tolerated toxicity and prolonged PFS in patients (Table 2, also see [179] for an early review). In 2012, a milestone phase III study (BOLERO-2) performed by Baselga et al. proved that everolimus (a rapalog, 10 mg daily) combined with exemestane (an AI, 25 mg daily) could improve PFS in ER^+^ BC compared to exemestane treatment alone [180]. Accordingly, the FDA approved the use of this combinational therapy for ER^+^ BC treatment. However, a follow-up study of BOLERO-2 failed to demonstrate that everolimus plus exemestane improved PFS [181]. Another similar combination, temsirolimus (rapalog) plus letrozole (AI), also failed to provide therapeutic benefits to BC patients [182]. Although a recent study showed that the addition of everolimus to fulvestrant (a selective ER degrader) treatment prolonged PFS in HR^+^/EGFR2^−^ patients, this was associated with an increase in the occurrence of adverse events [183]. Accordingly, interest in rapalogs as a treatment for BC patients has waned considerably and they are no longer being tested in clinical trials. 

Trials of rapalogs in PC, primarily in the CRPC setting, have also suffered setbacks recently (Table 2). Despite modest improvement of PFS in PTEN-null CRPC patients with everolimus treatment alone [187], most of the rapalog monotherapy regimens failed to show therapeutic benefits in PC patients [186,191,192], perhaps due to the fact that rapamycin can stimulate AR’s transcriptional activity [120]. A combination of everolimus and the AR antagonist bicalutamide in a phase II trial was unable to reach the primary endpoint due to low anti-tumour activity [185]. Similarly, temsirolimus and bevacizumab (vascular endothelial growth factor A inhibitor) treatment together not only failed to provide any improvement in terms of clinical outcome but also produced severe side effects in patients [192]. Moreover, two trials involving the combination of dual PI3K/mTOR inhibitor BEZ235 with abiraterone were also withdrawn during the early stages due to high levels of toxicity [188,189]. It has been well known that rapamycin does not inhibit mTORC1 completely [193] and may not affect mTORC2 [121], whereas the ATP-competitive mTORis can simultaneously and completely blunt the activation of wild-type mTOR-associated mTORC1 and 2 [194]. INK128, an ATP-competitive mTORi, was initially identified as a strong candidate for PC treatment due to prominent anti-tumour activity against PC progression in the cPTEN^fl/fl^ GEMM [156]. Unfortunately, a recent study found that INK128 exerted strong side-effects (grade 3 dyspnea and maculopapular rash) in CRPC patients, and thus the trial was also forced to be discontinued [190].

Some recent studies have unravelled mechanisms behind the resistance of BC/PC to mTOR inhibition. Reactivation of the PI3K/AKT pathway is a classic feedback mechanism driven by the rapalogs [83,122]. Yang et al. discovered that the induction of PI3K/AKT stimulation by everolimus in ER^+^ BC cells was mediated by the ER, which also required the transactivation of insulin receptor/IGF1R and insulin receptor substrates [195]. Additionally, screening of secreted proteins from the ER^+^ BC tumour microenvironment has revealed that the induction of fibroblast growth factor 2 was primarily responsible for conferring resistance to the BC cells against PI3K/mTORis [196]. 

In short, lessons from clinical trials of mTORis in BC and PC and associated mechanistic studies have established a challenging paradigm: the first generation of mTORis usually do not provide further clinical benefits on top of existing anti-BC/PC therapies, likely because they are relatively “weak” inhibitors against the mTORCs [193]; whereas “stronger” drugs such as the dual PI3K/mTOR or the ATP-competitive inhibitors are often associated with severe adverse effects on patients, since they also affect physiological functions of non-malignant cells [197] and may trigger feedback mechanisms to activate other oncogenic pathways [83]. The question of how to leverage mTORis such that significant anti-tumour activity is achieved with tolerable side effects remains outstanding.

### 7.3. The Future of mTOR Inhibition as a Therapy for Breast and Prostate Cancer 

Despite some unfavourable outcomes from those mTORi-related trials, more recent studies provide new hope. For example, fulvestrant-resistant ER^+^/HER2 kinase domain mutant (i.e., G309A, L755S and V777L) cells possess high level of PI3K/mTORC1 activity, as evidenced by increased levels of phosphorylated AKT and RP S6, respectively, and Everolimus was able to resensitize those cells to fulvestrant or estrogen deprivation [198]. In addition, an alternative approach would be to test other mTORis, especially beyond the rapalogs. GDC-0084, a dual PI3K/mTOR inhibitor designed to cross the blood-brain barrier [199], was recently shown to exert growth inhibitory effects on BC brain metastatic cell lines bearing *PIK3CA* mutations both in vitro and in vivo [200]. Earlier this year, La Manna et al. also reported the first attempt of using RapaLink-1 against PC; exposure of two bone-metastatic PC PDX models to RapaLink-1 strongly reduced their cancer stem cell features and prevented tumour growth in vivo [201]. However, although results from these preclinical studies appear promising, patients receiving these agents are more likely to experience stronger side effects than those who were administered with the rapalogs (see Section 7.2). Therefore, when it comes to future clinical trials, it is of utmost importance to ensure the safety aspect of these drugs does not compromise their potential as novel treatments.

## 8. Conclusions, Future Perspectives and Outstanding Questions 

While ER and AR are clear drivers of neoplastic growth, our understanding of their involvement in reshaping the translatome during cancer progression remains incomplete. We propose that the most pressing outstanding questions for basic science and for clinical translation are:-How is translation modulated during adaption to anti-estrogen/androgen therapies and in response to direct alterations to AR/ER (e.g., mutations and truncations)?-Why do different ER/AR ligands exert distinct downstream effects on protein synthesis?-How do cancer cells balance the need for increased protein synthesis while avoiding UPR?-Will single-cell technologies address key mechanistic questions related to the interplay between AR/ER signalling and protein synthesis?-Does the cytotoxicity of drugs that target the translation modulators (e.g., mTORis) primarily rely on their effect on protein synthesis?-How do we identify BC/PC patients who are likely to benefit from protein synthesis-targeted therapies? Can we use translation-related genetic profiles to tailor personalized treatment regimens? For example, patients harbouring PTEN/PI3K/mTOR mutations would be expected to gain the most benefit from PI3K/mTOR inhibitors.

## Figures and Tables

**Figure 1 cancers-13-03254-f001:**
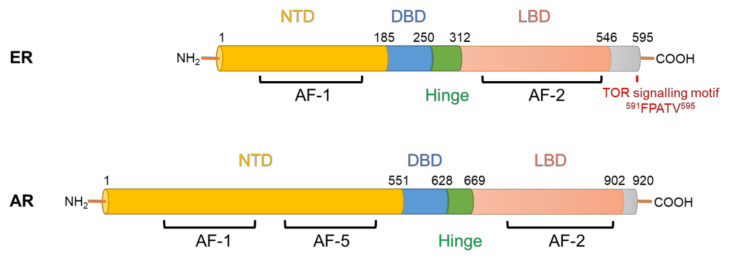
Structural domains of human ER and AR proteins. Both ER and AR belong to the nuclear receptor superfamily and share the same functional domains including the N-terminal domain (NTD), a DNA-binding domain (DBD), a hinge region and a ligand-binding domain (LBD). Activation function (AF) domains (AF-1, AF-2 and AF-5, the latter present only in AR) are shown.

**Figure 2 cancers-13-03254-f002:**
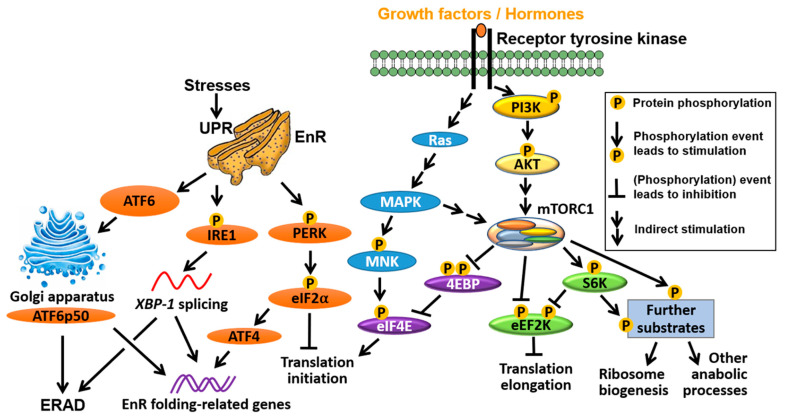
Signalling pathways that control translation. In response to extracellular stimuli, oncogenic signalling pathways, such as Ras/MAPK/MNK and PI3K/PKB, are activated and subsequently switch on/off downstream signalling events including mTORC1/S6K/eEF2K, mTORC1/4EBP/eIF4E and MNK/eIF4E pathways that drive/prevent translation initiation of elongation. In contrast, stress signals primarily act through three UPR pathways (PERK/eIF2α/ATF4; IRE1-*XBP1*; ATF6/ATF6p50) to restore protein homeostasis, which is achieved by blocking translation initiation (PERK/eIF2α), inducing the transcription of EnR folding-related genes (all three arms) and ERAD (through IRE1-*XBP1* and ATF6/ATF6p50).

**Figure 3 cancers-13-03254-f003:**
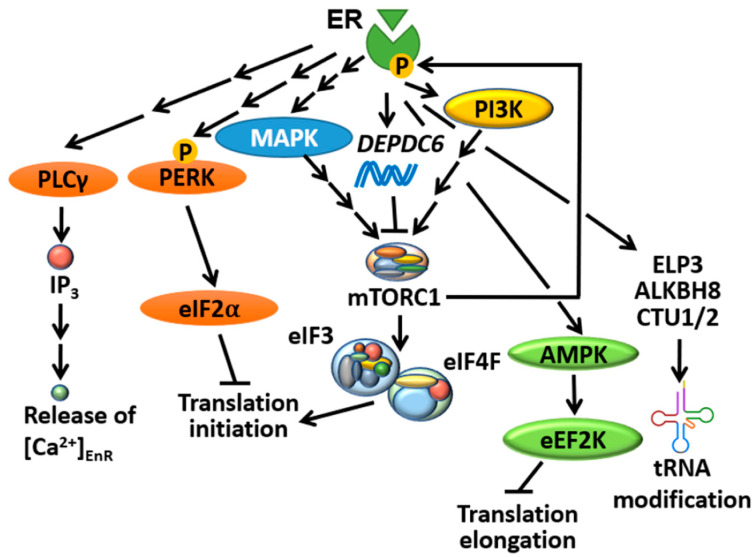
Schematic presentation summarizes reported major ER-driven translational control. ER can modulate translation initiation via stimulation of mTORC1 and eIF2α. It can also affect UPR via regulating the release of EnR Ca^2+^. ER inhibition triggers the AMPK/eEF2K pathway, and subsequently alleviates translation elongation. ER may also indirectly impact on translation elongation of mRNAs enriched in codons containing mcm^5^s^2^U^34^-modified tRNAs.

**Figure 4 cancers-13-03254-f004:**
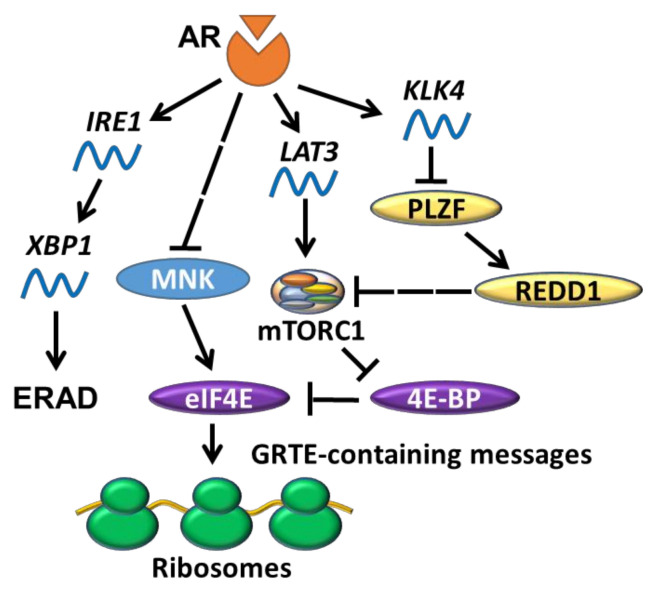
Signalling pathways implicated in AR-driven translational control. AR has been shown to modulate translation initiation through signalling cues (MNK and mTORC1, the latter via LAT3 and KLK4) that converge on eIF4E, a translation initiation factor that promotes the translation of mRNAs with certain patterns (e.g., guanine-rich translational element (GRTE)-containing messages). AR can also act as a regulator of the UPR by directly inducing *IRE1*, which leads to upregulation of ERAD-related genes.

**Table 1 cancers-13-03254-t001:** Frequency of mutations in selected translation-related genes encoding proteins upstream of mTORC1 in BC/PC according to cBioPortal (TCGA PanCan 2018; cbioportal.org).

Gene	BC (Invasive Carcinoma, 1084 Samples)		PC (494 Samples)	
	No. of Mutated Samples	% of Mutated Samples	No. of Mutated Samples	% of Mutated Samples
*AKT1*	27	2.49%	2	0.40%
*AKT2*	4	0.37%	0	0.00%
*AKT3*	8	0.74%	1	0.20%
*BRAF*	7	0.65%	7	1.42%
*CRAF*	7	0.65%	0	0.00%
*HRAS*	5	0.46%	4	0.81%
*KRAS*	6	0.55%	2	0.40%
*MAP2K4*	7	0.65%	1	0.20%
*MAP3K1*	7	0.65%	1	0.20%
*MTOR*	20	1.85%	2	0.40%
*PIK3CA*	333	30.72%	10	2.02%
*PIK3CB*	10	0.92%	3	0.61%
*PIK3CD*	11	1.01%	2	0.40%
*PTEN*	56	5.17%	13	2.63%

**Table 2 cancers-13-03254-t002:** A selection of reported clinical trials related to mTORis and their respective outcomes in BC/PC (chronological order). Abbreviations not defined in the main text: DI: dual PI3K/mTOR inhibitor; FA: FDA approval; Gen.: generation/class of mTORi (first listed under the column of “Drug(s)”); Ref.: reference.

Year	Cancer Type	Drug(s)	Gen.	Phase	Outcome Summary	Ref.
2005	Localized or metastatic BC	Temsirolimus (CCI-779) only	1st	II	Patients treated with temsirolimus showed anti-tumour activity and well tolerated toxicity.	[176]
2008	Advanced BC	Everolimus (RAD001) and letrozole	1st	I	The combinational therapy showed anti-tumour activity, toxicity also well tolerated.	[177]
2009	Recurrent or metastatic BC	Everolimus only	1st	II	Continuous daily dosing but not weekly dosing had anti-tumour activity. The drug was well tolerated but some patients developed pneumonitis.	[178]
2010	Localized PC	Sirolimus (rapamycin) only	1st	I	Daily dosing had no effect on tumour proliferation/apoptosis despite suppresion of RP S6 phsophorylation.	[184]
2012	CRPC	Everolimus and bicalutamide	1st	II	Combinational therapy was well tolerated despite 56% cases of grade 1/2 mucositis, but it had low activity and did not achieve the primary endpoint.	[185]
2012	HR^+^ BC	Everolimus and exemestane	1st	III, FA	Combinational therapy improved PFS of patients previously treated with non-steroidal AIs.	[180]
2013	Locally advanced or metastatic BC	Temsirolimus and letrozole	1st	III	In comparison to lotrozole monotherapy, daily and orally administrated temsirolimus failed to confer added PFS benefit to aromatase inhibitor-resistant ER^+^ patients.	[182]
2013	CRPC	Temsirolimus only	1st	II	Weekly dosing of the drug had minimal therapeutic activity, and the study was put on halt at an early stage.	[186]
2013	CRPC	Everolimus only	1st	II	The monotherapy regime modestly improved PFS, especially in PTEN^−/−^ patients. Toxicity was also manageable.	[187]
2014	Advanced BC	Everolimus and exemestane	1st	III	Addition of everolimus to exemestane treatment did not provide further improvement to HR^+^ BC patients at the secondary endpoint.	[181]
2017	CRPC	BEZ235 with abiraterone/prednisone	DI	I	The combinational therapy was poorly tolerated; adverse effects included mucositis, hypotension, dyspnea and pneumonitis.	[188]
2017	CRPC	BEZ235/BKM120 with abiraterone	DI	Ib	The trial was discontinued due to high levels of toxicity and poor pharmacokinetics among the participants.	[189]
2018	CRPC	MLN0128 (INK128)	2nd	II	MLN0128 exhibited high levels of toxicity in patients; dyspnea and maculopapular rash were the main grade 3 adverse events.	[190]
2018	ER^+^/EGFR2^−^BC	Everolimus and fulvestrant	1st	II	Despite increased cases of adverse events, improved PFS was observed with combinational therapy-treated patients.	[183]
2019	High-risk localized PC	Everolimus	1st	II	Everolimus showed limited clinical activity.	[191]
2019	CRPC	Temsirolimus and bevacizumab	1st	I/II	The combinational therapy did not improve the clinical outcome of the participants, and also induced severe adverse events.	[192]

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
