# Peer review of "Regulation of mRNA Translation by Hormone Receptors in Breast and Prostate Cancer"

_cancers, 2021, doi:10.3390/cancers13133254_

Round 1

Reviewer 1 Report

In the submitted manuscript Xie et al. gave an extensive overview of the causes of protein translation (dys)regulation by estrogen and androgen receptors in breast and prostate cancers, respectively, and that associated therapeutic approaches. This review paper would perfectly fulfill in scientific literature often unfairly underrepresented the roles of protein synthesis dysregulation as a molecular culprit of cancer.

I have several, mostly stylistic, comments and suggestions:

1) IMHO, BCa and PCa are kind of strange choice for abbreviations for breast and prostate cancer, respectively, since BC and PC are not used in any other context in the text, while CRPC and not CRPCa is used as an acronym for castration-resistant prostate cancer.

2) I strongly suggest authors to consistently use the HGNC (https://www.genenames.org/) approved gene and protein names and symbols; to avoid hyphens and Greek letters, and rather use e.g. 'alpha' or just 'A'; to consistently write symbols of eIFs' subunits in capital letters (please pay special attention to distinguish EIF2A and EIF2S1); to avoid abbreviations like PtdIns, to check throughout the manuscript that all abbreviations are explained after first mentioning (e.g. PDX, TNBC, ADT...).

3) This extensive review would greatly benefit from the list of abbreviations, especially figures, since not all abbreviations presented on them are explained in figure legends.

4) Line 30: Since prostate is not paired organ, I would prefer singular: "Breasts and prostate are...".

5) In Table 1 you presented frequencies of mutated  BC/PC patient samples for a selection of translation-related gene. Likewise, it would be very interesting to make a similar figure showing the expression profiles of translation-related genes in BC/PC compared to "normal" (healthy) controls (e.g. using GEPIA or any other similar tool).

6) In lines 200-201 you correctly stated "Such extra-nuclear functions of nuclear receptors have been proposed to allow more rapid responses than their transcriptional effects." However, it is often the case of completely different class of membrane sex hormone receptors, like GPER, ER-X, ERx, Gq-mER as mERs, and GPRC6A and ZIP9 as mARs.

7) Line 209: Those letters do not need to be underlined.

8) Line 265: If it is the case of human BRAF protein, its name should be italicized.

9) Line 268: Word "that" is duplicated.

10) Line 336: Construction "metastatic castration-resistant PCa (CRPC) (mCRPC)" is kind of awkward.

11) What about non-canonical mechanisms of protein translation initiation? It would be interesting and informative to discuss also about the relations between ER/AR and cap-independent, i.e. IRES-dependent translation initiation, since its role in carcinogenesis, as well as potential thepeutic approaches, are interesting, but unfortunately also understudied/under-reported.

Author Response

Please see attached file for our responses to the reviewer's comments.

Reviewer 2 Report

The review entitled is greatly interesting and will be appreciated by our scientific community.

However, some points should be clarified for our readers.

  • Lane 15 : It is mentioned « estrogen receptor-α (ER) ». So, authors should keep this abbreviation (ER) in all the manuscript, as sometime ERα is mentioned (Figure 1 and its legend, lanes 34, 38, 56, 205, and 297 for example) instead. Also, it will be interested in this review concerning mRNA translation by hormone receptors to precise the potential contribution of ER-beta (ERß).

  • Lane 65 : AE should be replaced by ER

  • The authors made a great effort to give a definition for abbreviations. However, some remain undefined as AR-FL and AR-V7 (lane 450), Esr1(lane 267), HR (Lane 587), PDXs (lane 336), PFS (lane 281). Moreover, AR-V7 need to be introduced. 

  • Lane 317 : SERCA need to be introduced.

  • It would be fine if the location of the highly conserved TOR signalling motif (FPATV), mentioned lane 272, is annotated on ER protein representation (Figure 1).

  • Lane 284 : « … metastatic ER+ BCa cell lines harbouring the 8p11-p12 amplicon » : Please introduce this amplicon and discuss the potential effect of TACC1-EIF4EBP1gene fusion.

  • Some typing errors, such as doubling dots in figure 1 legend and word spacing, are remaining in all the text.

Author Response

(The authors gave the same response as above.)
